# Health Risk Assessment for Human Exposure to Heavy Metals via Food Consumption in Inhabitants of Middle Basin of the Atrato River in the Colombian Pacific

**DOI:** 10.3390/ijerph20010435

**Published:** 2022-12-27

**Authors:** Gabriel Caicedo-Rivas, Manuel Salas-Moreno, José Marrugo-Negrete

**Affiliations:** 1Biosistematic Research Group, Biology Department, Faculty of Natural Sciences, Universidad Tecnológica Del Chocó, Quibdó 270002, Chocó, Colombia; 2Faculty of Basic Sciences, Universidad de Córdoba, Carrera 6 No. 76-103, Montería 230002, Córdoba, Colombia

**Keywords:** vegetables, fruits, fish, tubers, MeHg, As

## Abstract

The Atrato river basin is one of the world’s most biodiverse areas; however, it is highly impacted by mercury gold mining, which generates air, water, and soil pollution. (1) Background: The concentrations of persistent heavy metal pollutants, mercury (Hg), lead (Pb), cadmium (Cd), and arsenic (As) in the fish, fruits, and vegetables most consumed by the riverside inhabitants of the middle basin of the Atrato river represent a danger to public health; (2) Methods: A total of 154 samples of different fruits and vegetables and 440 samples of fish were analyzed by atomic absorption spectroscopy. A sample of 446 people were surveyed to evaluate food consumption and carcinogenic and non-carcinogenic risk; (4) Conclusions: High concentrations of As, Hg, Pb, and Cd were identified in fish, fruits-tubers, and vegetables-stems commonly consumed by inhabitants of the middle basin of the Atrato River, which exceeded the Codex limits and the limits established by the WHO/FAO, especially for carnivorous fish species. A high carcinogenic and non-carcinogenic risk was evidenced amongst inhabitants of the middle basin of the Atrato River due to the consumption of fish contaminated with high concentrations of As, MeHg, and THg. The risk due to the consumption of vegetables was very low.

## 1. Introduction

The contamination of soil, air and water by heavy metals is a global concern due to its ability to affect different biological systems. Heavy metal contamination can result in bioaccumulation and biomagnification in the food chain affecting human health. Metal contamination in aquatic and terrestrial ecosystems is a serious environmental problem. Elements such as lead (Pb), arsenic (As), cadmium (Cd), mercury (Hg) and the organic form of the latter, methylmercury (MeHg), are toxic, even at low concentrations, and have a high capacity for bioaccumulation. The US Agency for Toxic Substances and Disease Registry (ATSDR) ranks these metals high on its 2017 Priority List of Hazardous Substances [1]. Contamination by these heavy metals can have various origins. However, this type of contamination is generally associated with anthropogenic activities, including gold mining, specifically the extraction of the precious metal by amalgamation with Hg. This activity generates significant levels of Hg contamination in soil and water; reports indicate that this type of activity produces emissions equivalent to about 880 tons of Hg per year [2,3,4]. A worrying aspect of these activities is that they generate heavy metals, such as Cd, Pb, and As, which are associated with gold minerals. These can be dispersed through erosion and chemical weathering of tailings from gold mining [5]. Some 200 tons of Hg are used in gold mining activities in Colombia, which generate emissions that range between 30 and 70 tons of Hg [6,7]. In the Chocó department, gold mining has been an important socioeconomic pillar in many communities for many years; it represents the base of the economy in many places within the department; however, these activities have generated environmental impacts on important water sources, such as the Atrato River [8]. Significant amounts of Hg are used in gold mining—approximately 150 tons—in this way a large amount of mining waste enters the Atrato river basin or its tributaries, contaminating the water and sediments [7,9]. In the sediments of the river basin, Hg is transformed into MeHg by the action of bacteria. This organic form of Hg is quite toxic—it can cross membranes and bioaccumulate in freshwater fish species in toxic concentrations [10,11]. The incorporation of heavy metals and MeHg into the food chain is one of the biggest concerns of these activities due to the inevitable transfer to human beings, with food intake as the main route of exposure. According to WHO reports, approximately 500,000 people perish annually across the world due to the consumption of contaminated food, 80,000 of which are associated with Colombia [12]. For this reason, the international standards that regulate the limits of tolerance or acceptance of the levels of heavy metals in food are important, with the objective of protecting human health in each region [13].

There have been several reports on different continents related to the risk assessments or quantification of heavy metals in plant and/or animal material arising from contamination as a direct result of the intervention of mining [14,15,16,17]. In Colombia, some investigations have reported high concentrations of these metals in fish commonly consumed in various communities; in addition, assessments of risk associated with the consumption of fish contaminated by heavy metals have been undertaken [18,19]. In the Colombian Pacific, Hg, Cd, Pb, and MeHg in fish from the Atrato river basin have been reported in high concentrations, particularly in fish with carnivorous habits [8,9,20,21]. In addition, Salazar-Camacho et al. [8] carried out a risk assessment of the consumption of fish in relation to levels of Hg, Cd, Pb, and MeHg in the riverside population of the Atrato River basin. It is important to note that most of the food consumed by families in the municipalities of the riverside areas of the Atrato River basin comes from crops and fishing in the territory itself. For this reason, the consumption of contaminated fish, fruits and vegetables is considered the main source of human exposure to heavy metals [22]. Therefore, the objectives of this study were: (1) to determine the concentrations of Hg, Cd, Pb and MeHg in fish, fruits and vegetables; (2) to determine the target hazard quotient (THQ) and total THQ to assess the carcinogenic and non-carcinogenic risk of metals from fish consumption; and (3) to evaluate the risk to human health due to the consumption of fish and vegetables (fruits-tubers and vegetables-stems) contaminated with heavy metals in residents of riverside municipalities of the middle basin of the Atrato River.

## 2. Materials and Methods

### 2.1. Study Area

This research was carried out in the Atrato river basin, a biodiverse lotic ecosystem located in the Colombian Pacific (Figure 1). It is made up of large bodies of water, forests, wetland swamps, grasslands, vast expanses of land used for agriculture, and many rural communities [23]. The basin has a depth of 31–38 m, an area of 35,700–36,400 km^2^, a length of 750 km, and a width that varies between 150 and 500 m. It arises in the municipality of El Carmen de Atrato, specifically in the Cerro Plateado, and empties into the Gulf of Urabá, in the Caribbean Sea. The basin has a flow of approximately 4137 m^3^/s, an annual precipitation between 5000- and 12,000- mm year^−1^, and an average annual temperature of 26 °C [23,24]. It receives the flow of water from many rivers along its course, some of which have high levels of mining waste contaminated with Hg, Cd, Pb, and As. Among the most important of these are the Quito, Bebaramá, Bebará, Neguá, and Cabí [25,26].

Our study was conducted in four areas of the tropical geographic basin of the Atrato River. The areas were the Medio Atrato, Bojaya, Murindó and Vigía del Fuerte, which were selected because they are important sites of gold mining [25,27]. The study was carried out on highly consumed fish, fruits and vegetables in the study areas. The fish samples were captured in the Atrato River within the study areas. Fruits and vegetables (fruits-tubers and vegetables-stems) are cultivated by the inhabitants of the communities for self-consumption and commercialization. These plants are cultivated in many cases in soils contaminated with heavy metals or, often, are treated with irrigation systems with water from the Atrato River.

### 2.2. Sampling

Fish: The capture of ichthyological material involved guides from each sampling point using artisanal fishing equipment, such as fishing nets, cast nets, and fishing rods. In total 440 individuals of 19 fish species were collected. The carnivorous species (330 individuals) were *Argeneiosus pardalis*, *Pimelodella chagresi*, *Rhamdia quelen*, *Trachelyopterus fisheri*, *Pimelodus punctatus*, *Andinoacara pulche*, *Leporinus muyscorum*, *Caquetaia kraussii*, *Caquetaia kraussii*, *Caquetaia kraussii*, *Caquetaia kraussii*, *Caquetaia kraussii*, and *Pimelodus punctatus*, *Leporinus muyscorum*, *Caquetaia kraussii*, *Ctenolucius beani*, *Pseudopimelodus schultzi*, *Sternopygus aequilabiatus*, *Hoplias malabaricus*, *Astyanax fasciatus*, *Caquetaia umbrifera*, *Cynopotamus atratoensis*, *Geophagus Pellegrini*, and *Pimelodus* sp. The non-carnivorous (110 individuals) were *Prochilodus magdalenae* and *Hypostomus hondae*. The fish samples were placed in polyethylene bags which had been previously labeled, placed in a polystyrene cooler to conserve the specimens, and transported to the Toxicology and Environmental Management laboratory of the University of Córdoba (Colombia). Subsequently, the total length of each fish was measured and, through an incision in the dorsal muscle, ten grams of tissue were extracted in one portion and again kept cold (4 °C) until the concentrations of the metals under study and the percentages of methylmercury (MeHg) were determined.

The fish species were identified using specialized taxonomic keys [28] with the help of the ichthyology team of the Technological University of Chocó (Colombia) and field guides.

Plants: The collection of vegetable material was carried out with the help of field guides at each sampling point. A total of 154 individual samples were collected and classified into two groups, fruits (12 species) and stems (one species). The fruit species were *Alibertia patinoi*, *Cocos nucifera*, *Citrus aurantifolia*, *Solanum sessilliflorum* Dunal, *Zea mays*, *Carica papaya*, *Musa balbisiana*, *Musa × paradisiac*, *Oryza sativa*, *Musa sapientum*, *Musa* sp., *Dioscorea trifida*; tubers (2): *Colocasia esculenta*, *Manihot esculenta*; vegetables (9): *Eryngium foetidum*, *Ocimum basilicum* L., *Ocimum tenuiflorum*, *Allium fistulosum* L., *Ocimum campechianum*, *Basella rubra* var, *Zingiber officinale*, *Origanum vulgare*, *Minthostachys mollis*. The stem species was *Saccharum officinalis*. The collections were georeferenced in situ, deposited in polyethylene bags and transported to the laboratory of Toxicology and Environmental Management of the University of Córdoba (Colombia). There, maceration was carried out followed by cold storage until the concentrations of the metals under study were determined.

The species were identified using specialized taxonomic keys [29] with the support of the biosystematics team of the Universidad Tecnológica del Chocó, the Chocó herbarium, the Universidad de Córdoba, and field description notebooks.

### 2.3. Analysis of THg, MeHg, As, Pb, and Cd in Fish Muscle, Fruits, and Vegetables

Quantities of 0.02 g of freeze-dried fish and plant material samples were analyzed for Hg concentration levels by atomic absorption spectrometry using a direct mercury analyzer (DMA-80 TRICELL, Milestone Inc, Italy) using the established EPA Method 7473. (EPA, 1998). For Cd and Pb analysis, Method 3051 A [30] and the procedure described by Karadede and ÜnlÜ [31] were used, respectively. The samples were digested with HNO_3_/HCl (1:3 *v*/*v*) and Cd and Pb analyses were performed using a Thermo Elemental Solaar S4-graphite furnace method. Analysis was carried out by calcining a mixture of 1 g of each fish and plant material sample with Mg (NO_3_) 2 at 550 °C in a muffle furnace, then 1 mL of concentrated HNO_3_ was added and heated to dryness, subsequently dissolved with 4.5 N HCl, filtered through a 0.45 µm filter, then topped up to 25 mL with distilled water (Szkoda et al., 2006). A Thermo Scientific iCETM 3500 AAS atomic absorption spectrometer, coupled to a VP100 continuous flow steam generator (Waltham, MA, USA), was used for As analysis (HGAAS; standard Methods SM 31114, 2017). Certified µreference materials (CRM) IAEA 407 and DORM-4 and triplicate evaluation were used for quality control of the methods used. The recovery percentage was between 92 and 96% and the detection limits for the different metals were 0.014 μg g^−1^ for Hg, 0.006 μg g^−1^ for Cd, 0.010 μg g^−1^ for Pb, and 0.016 μg g^−1^ for As. MeHg analysis was only performed on the fish samples; for MeHg quantification, approximately 0.2–0.3 g of fresh fish were digested with hydrobromic acid and toluene. The resulting mixture was centrifuged and extracted several times with L-cysteine. Finally, a 100 μL aliquot of the aqueous phase was injected into DMA [32]. Quality control of the method was performed in triplicate using a CRM DORM-2 standard of dogfish muscle certificate (4.47 ± 0.32 μg g^−1^). The percent recovery for MeHg was 99%. The limit of detection was 0.007 μg g^−1^, while the limit of quantification was 0.023 μg g^−1^.

### 2.4. Estimated Daily Intake (EDI)

A risk assessment of the estimated daily intake of fish, fruit, and vegetables in the municipalities of the middle Atrato basin was calculated using factors such as food consumption (µg kg^−1^ (bw) week^−1^), the concentration of the metal, and the body weight (bw), necessary for the determination of the estimated daily intake (EDI). The average body weight in adults living in the middle basin of the Atrato river was 69.2 ± 3.3 kg (Appendix A). On average, the inhabitants of the middle basin of the Atrato river consume 256 g/day of fish [8]. The average consumption of fruits and vegetables was established individually for each food; measurements were made by weighing the food portions from the information provided in the surveys (Appendix A). The results obtained from the analysis of the concentrations of Hg, As, Cd, Pb, and MeHg in µg/kg of wet weight, and from the surveys, made it possible to calculate the necessary parameters to evaluate the risk of human exposure to these contaminants by the consumption of fish, fruits, and vegetables. For this calculation, the equation described by Chien et al. [33] was used.
(1)$$\mathrm{EDI}=\frac{C\times \mathrm{Cconc}}{\mathrm{BW}}$$

### 2.5. Determination of the Target Hazard Quotient (THQ)

The non-carcinogenic risk was calculated using the THQ formula (HQ/RfDo). When the THQ is less than one, it indicates that the hazard quotient (HQ) is below the reference dose (RfDo) and, therefore, that daily exposure at this level is unlikely to cause adverse effects over a person’s lifetime. The THQ is a calculation using the assumptions of the US EPA Integrated Risk Analysis (USEPA, 2000). The THQ was determined using the following equation [34]:(2)$$\mathrm{THQ}=\frac{\mathrm{EFr}\times \mathrm{EDtot}\times \mathrm{FIR}}{\mathrm{RfDo}\times \mathrm{Bw}\times \mathrm{ATn}}\times C-10^{-3}$$
where EDtot is the exposure duration (30 years), EFr is the exposure frequency (350 days/year), FIR is the food ingestion rate (g/day), 10^−3^ is the unit conversion factor (kg/g); C is the element concentration in fish (μg/g ww), RfDo is the oral reference dose (mg/kg-day), Bw is the average adult body weight according to surveys in each municipality and ATn is the average exposure time for non-carcinogens (365 days/year × number of exposure years, assuming 30 years). The total THQ (TTHQ) was expressed as the sum of the THQ values for each studied element [34]:

Total TQH (TTHQ) = TQH(Toxican 1) + TQH(Toxican 2) + TQH(Toxican 3) + …
(3)


### 2.6. Carcinogenic Risk Assessment (CR)

The contaminants associated with carcinogenic risk are As and Pb. The CR is defined as the lifetime chance of an individual developing any type of cancer due to exposure to carcinogenic hazards [35,36]. These carcinogenic health risks are calculated individually for each element throughout its lifetime according to the following equation [37,38]:(4)$$\mathrm{CDDingestion}=\frac{C\times \mathrm{FIRing}\times \mathrm{ED}\times \mathrm{EF}}{\mathrm{Bw}\times \mathrm{AT}\times \mathrm{SF}}$$

CR = CDDingestion × SF
(5)

where CDDingestion is the chronic daily dose (mg/kg/day) established for potentially toxic heavy metals received by ingestion; C is the heavy metal content in fish and vegetables (µg/g); FIRingestion is the ingestion rate: 256 mg/day in fish [8], in vegetables the quantity varies (Appendix A)—these amounts are for adult women and men who lived in the middle zone of the Atrato river basin; ED is the exposure duration, six years for children and 30 years for adults [35]; EF is the exposure frequency—in this study, 365 days/year; SF is the slope factor (kg/mg/Day)—SF is 1.5 for As and 0.0085 for Pb [38]; Bw is the mean body weight for each municipality; AT is the mean time for carcinogens (As and Pb) 70 × 365 days [38]; and CR is the carcinogenic risk—when the CR value is less than 1 × 10^−6^, the risk is regarded as negligible, and if the CR value exceeds 1 × 10^−4^, there is likely to be a risk to human health [37].

### 2.7. Assessment of Human Health Risk Related to MeHg

The risk assessment was carried out with a total of 769 voluntary respondents surveyed. Data was collected on educational level, average body weight, gender, frequency of fish consumption per week, number of times each participant ate fish per day in a week, and the type of fish. A total of 323 respondents were male and 446 were female. All respondents were aged ≥ 15 years. The respondent sample was divided into two groups: the first group, comprising children and women of childbearing age (WCHA) and the second group comprising the rest of the adult population (GP) (Appendix A). The potential risk of human exposure to MeHg was assessed according to the estimated weekly intake (EWI-μg/bw/week) using the equation described by UNEP [39]:(6)$$\mathrm{EWI}=\frac{\mathrm{IR}\times C}{\mathrm{Bw}}$$
where IR is the weekly intake (g/week) of fish, C is the median concentration of MeHg (μg/kg) in fish, and Bw is the bodyweight of the person (kg). The IRs were calculated taking into account the consumed portion of fish (g/day) and the frequency of consumption (days/week) in the four municipalities of the Atrato river basin.

The concentration of MeHg that the consumed fish species should contain to avoid exceeding the provisional tolerable weekly intake (PTWI) [34] was calculated using the following equation:(7)$$[\mathrm{MeHg}]\mathrm{permissible}=\frac{C\times \mathrm{PTWI}}{\mathrm{EDI}}$$
where PTWI is the reference value of 1.6 μg/kg bw/week for women of childbearing age and children, 3.2 μg/kg bw/week is the reference value for the adult population [34] and C is the median concentration of MeHg (μg/kg) in fish. The amount of fish consumption is crucial in risk assessment as it plays a key role in the generation of adverse effects on human health. For this reason, we estimated the maximum quantity of fish that a person could consume weekly (MFW) without adverse health effects, according to the following equation:(8)$$\mathrm{MFW}=\frac{\mathrm{PTWI}\times \mathrm{IR}}{\mathrm{EWI}}$$

Finally, to calculate the degree of Hg contamination in the most consumed fish species, we used the formula proposed by Zhang et al. [40]:(9)$$\mathrm{Pi}=\frac{\mathrm{Ci}}{\mathrm{Si}}$$
where Ci and Si are the median concentration of the metal in the fish muscle and the value of the evaluation criteria, respectively, and Pi is the pollution index. Two reference limits were used: a threshold of 200 μg/kg ww [41] for vulnerable populations, such as children under 15 years of age and women of childbearing age, and a threshold of 500 μg/kg ww [42] for the adult population.

### 2.8. Data Analysis

Kolmogorov–Smirnov (n ≥ 50) and Shapiro–Wilk (n < 50) tests were used to assess whether data did or did not follow a normal distribution. The Kruskal–Wallis test was employed to evaluate the differences among Hg, As, Pb, and Cd concentrations between fish species. Spearman’s test was performed to evaluate the correlation between the concentrations of the elements and the trophic level of the fish. A *p*-value of 0.05 was chosen to indicate statistical significance. THg, MeHg, As, Pb, and Cd concentrations were expressed as μg/kg ww of fish, fruits and vegetables. The statistical analyzes were carried out using the R Project statistical program version 3.6.1 (R Core Team, Vienna, Austria).

## 3. Results

### 3.1. Concentrations of Hg, Cd, Pb, As, and MeHg in Fish, Fruits, and Vegetables

The concentrations of Hg, Pb, Cd and As (µg kg^−1^), the percentages of MeHg (%MeHg), the fish species consumed by the inhabitants and the trophic level of the fish species collected in the middle basin of the Atrato river are shown in Appendix A. Based on the taxonomic identification, 19 species of fish were obtained, with a total of 440 individuals. Of these species, five were carnivorous (59 individuals), two were detritivorous (110 individuals), six were omnivorous species with a tendency to carnivory (120 individuals), one was omnivorous (14 individuals), one was an omnivorous species with a preference for fish and plant material (26 individuals), and one was a piscivorous species (111 individuals). The most common species were *Prochilodus magdalenae* (15.9%), *Hoplias malabaricus* (11.1%), *Rhamdia quelen* (10.0%), *Hypostomus hondae* (8.9%), and *Astyanax fasciatus* (8.6%). The concentrations of Hg were highest, followed by those of As, Pb, and Cd. (Appendix A).

For the fish samples evaluated, the minimum values of Hg were found between two municipalities in Vigía del Fuerte, for *Leporinus muyscorum* with 44.5 ± 22.7 µg kg^−1^ and in Murindó for species *Hypostomus hondae* (41.5 ± 32.5 µg kg^−1^) and *Andinoacara pulcher* (32.9 ± 4.3 µg kg^−1^). Samples from the municipality of Vigía del Fuerte contained the highest concentration of As in *Ctenolucius beani* with 1008.0 ± 552.7 µg kg^−1^ (Appendix A). Of the total fish samples for Hg and MeHg, 221 individuals (piscivores 111, omnivores with a tendency to carnivory 74, and carnivores 36) exceeded the limit for populations at risk, which was established at 200 µg kg ^−1^ [41]. Among these, 102 individuals (53 piscivores, 26 omnivores with a tendency to carnivory, and 25 carnivores) exceeded the maximum recommended limit for human consumption established in 500 µg kg^−1^ [42]. The species *C. beani* and *Ageneiosus pardalis* exceeded the maximum permissible limits for THg 500 µg kg ^−1^. Some species also exceeded the maximum permissible limits: *A. pardalis* (Murindó and Vigía del Fuerte), *Trachelyopterus fisheri* (Murindó and Vigía del Fuerte), *H. malabaricus* (Murindó), *Cynopotamus atratoensis* (Vigía del Fuerte), *C. beani* (Murindó and Bojayá), *Sternopygus aequilabiatus* (Murindó and Vigía del Fuerte), *Caquetaia kraussii* and *R. quelen* (Murindó). Five species exceeded the threshold for MeHg 500 µg kg^−1^; these were: *A. pardalis* (Murindó and Vigía del Fuerte), *T. fisheri* (Murindó and Vigía del Fuerte), *C. beani* (Murindó and Bojayá), *C. atratoensis* (Vigía del Fuerte) and *S. aequilabiatus* (Vigía del Fuerte). *C. kraussii* and *H. malabaricus* were close to reaching this threshold, with concentrations of 473.05 and 485.71 µg kg^−1^, respectively (Figure 2a). The species *P. magdalenae* was the most consumed in the middle basin of the Atrato river; its concentrations of THg and MeHg did not exceed the established thresholds of 200 µg kg^−1^ and 500 µg kg^−1^.

None of the mean concentrations for the 19 species of fish reported in this study were higher than the maximum permissible levels of Cd, Pb, and As in the muscle of the fish, which were established at 300, 100, and 1000, µg kg^−1^, respectively (Figure 2c–e) [43,44]. For As, two species *Geophagus Pellegrini* and *H. hondae* presented concentrations > 100 µg kg^−1^ < 300 µg kg^−1^ (Figure 2c).

The four metals studied were quantified in vegetables for two groups: fruits and vegetables; the species of vegetables with a total of 24 species were subdivided into 15 fruits and 9 vegetables. A total of 154 individual specimens are listed in Appendix A. Hg presented the lowest concentration in the species *Citrus aurantifolia* for the municipalities of Murindó, Bojayá, and Vigía del Fuerte, with *O. vulgare* being the species with the highest mean concentration value (Figure 3a). The minimum values for As concentrations were in the species *Musa paradisiaca* and *Alibertia patinoi* (3.18–6.61 mg kg^−1^). The maximum mean concentrations of As were found in Murindó (60.84 ± 107.8 mg kg^−1^) and Bojayá (57.1 ± 99.6 mgkg^−1^). The species with the highest concentrations for As were *Minthostachys mollis* and *Ocimum campechianum*. The concentrations for As observed in the food samples indicated that three species contained high concentrations and exceeded the permissible WHO limit of 100 mg kg^−1^ [41]; these were *Origanum vulgare*, *A. patinoi,* and *O. campechianum* (Figure 3b). Pb showed the lowest average concentration levels by municipality in Murindó (4.3 ± 0.0 mg kg^−1^) and Bojayá (4.7 ± 1.5 mg kg^−1^). For Cd, none of the observed concentrations were at a level above that recommended by the WHO (100 mg kg^−1^) (Figure 3c). However, the mean concentrations of Hg, Pb, Cd, and As in fruits and vegetables in each municipality exceeded the maximum permissible levels of concentrations established by the Codex for Hg: 0.1 mg kg^−1^, Pb: 0.1 mg kg^−1^, Cd: 0.05 mg kg^−1^ and As: 0.1 mg kg^−1^ [45] and by the EU for Hg: 0.1 mg kg^−1^, Pb: 0.2 mg kg^−1^, Cd: 0.05 mg kg^−1^ and As: 0.2 mg kg^−1^ [46] (Appendix A).

### 3.2. Species of Fish, Fruits, and Vegetables Most Consumed in the Middle Basin of the Atrato River

The data obtained for the average consumption of fish, fruits, and vegetables amongst inhabitants of the municipalities of Medio Atrato, Bojayá, Vigía del Fuerte, and Murindó, belonging to the middle basin of the Atrato river, indicated that, of the 19 species of fish captured, nine were associated with a high preference for consumption amongst riverside inhabitants of this middle area of the basin (Appendix A). Among these, two species of non-carnivorous habits stood out: *P. magdalenae* and *H. hondae*. The results showed six species with high rates of population intake, including *P. magdalenae*, *L. muyscorum*, *P. schultzi*, *A. pardalis*, *H. Malabaricus* and *R. quelen* with 4.3, 4.3, 4.2, 3.5, 3.6 and 3.7 days/week reported mean consumption, respectively.

**Figure 2 ijerph-20-00435-f002:**
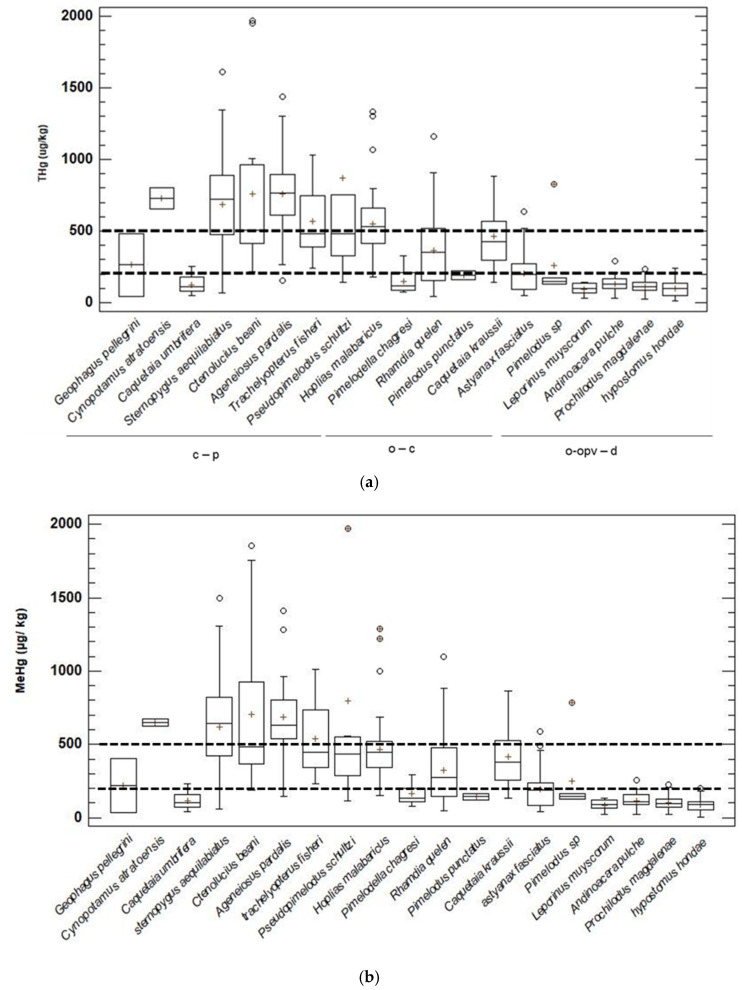
Concentrations of Hg (**a**), MeHg (**b**), As (**c**), Pb (**d**), and Cd (**e**) (μg kg^−1^ ww) in fish samples from the middle basin of the Atrato River. Consumption limits proposed by WHO [47,48] for these contaminants are as follows: for Hg and MeHg, threshold value of 500 ugkg^−1^ of body weight for the adult population; threshold value of 200 ugkg^−1^ of body weight for vulnerable populations (children, the elderly, and women of childbearing age). For As, Pb and Cd, 1000, 300 and 100 µg kg^−1^, respectively [49,50]. Feeding habits: c-carnivore, p-piscivore, o-omnivore, d-detritivore, oc-omnivore/carnivore and opv-omnivore/piscivore. Scattered concentrations (°), mean concentrations (+).

A total of 24 plant species were analyzed in the different municipalities studied, seven of which were associated with a high rate of preference for population consumption. Among the fruits, five stood out: *M. balbisiana*, *M. sapientum*, *C. aurantifolia*, *M. paradisiaca* and *O. sativa*; for vegetables, the data showed that only two species were associated with a high frequency of preference for population consumption: *E. foetidum* and *O. campechianum* (Figure 3).

**Figure 3 ijerph-20-00435-f003:**
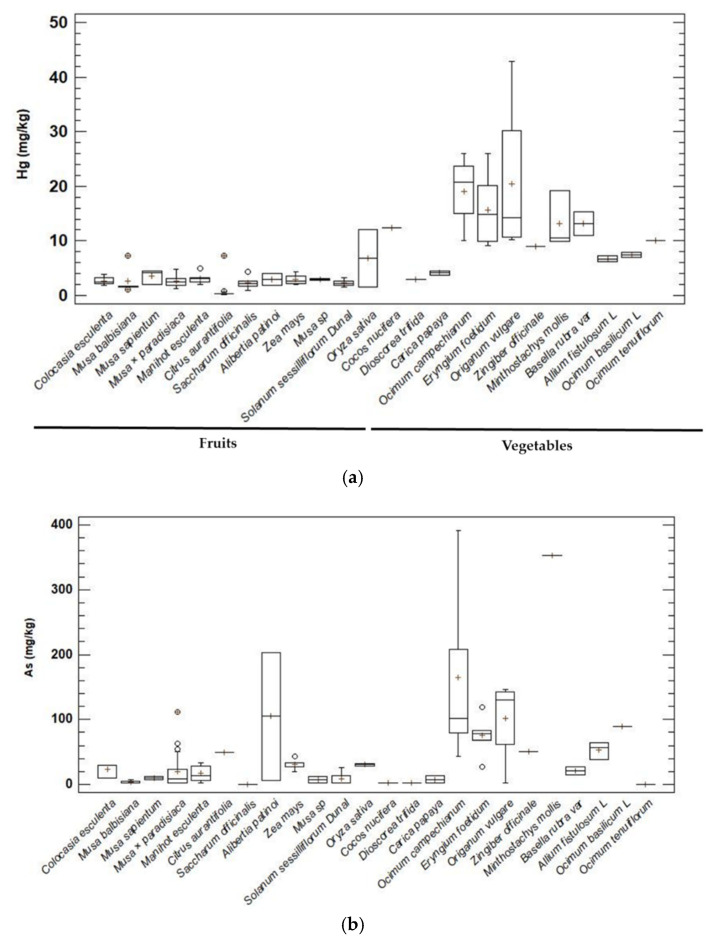
Concentrations of Hg (**a**), As (**b**), Pb (**c**), and Cd (**d**) in fruits and vegetables commonly consumed in municipalities of the middle basin of the Atrato River. Consumption limits proposed by the WHO: threshold value of 100 mg kg^−1^ of body weight for the adult population and for vulnerable populations (children, the elderly and women of childbearing age) for the heavy metals studied.

### 3.3. Determination of Human Health Risk by Fish and Vegetables Consumption

The general characteristics of the population of the middle basin of the Atrato River are described in Appendix A. A total of 57.9% (n = 446) of the respondents were women and 42.1% (n = 323) were men, with a mean age between 39 and 46 years (range: 15.2–88.8) and average weight between 68 and 72 kg.

The evaluation of the risk to health due to food consumption is important when considering the amounts consumed in the population under study. The average amount of fish consumed by inhabitants of the middle basin of the Atrato River was 256 g/day [8], with an average frequency of consumption of 3.4 days/week. The total weight of fish consumed was greater than 30 kg year^−1^.

For fruits and vegetables, the amount consumed varied according to the food. The average frequency of consumption of fish, fruits, and vegetables for inhabitants of the middle basin of the Atrato river was 3.4 days/week.

The risk arising from the consumption of fish, fruits, and vegetables was calculated using the EDI, THQ, TTHQ, and CR metal contamination indices to estimate the accumulation and risk levels of metals for the most consumed species (Table 1 and Table 2).

### 3.4. Determination of Health Risk from the Consumption of Fish, Fruits, and Vegetables

To assess the risks to human health based on frequent exposure of an individual through consumption of different species of vegetables and fish that accumulate different levels of contamination, the estimated daily intake (EDI) for each metal was determined. For the vegetables studied, the EDI values were 0.0135 μg/kg/day (range: 7.882 × 10^−5^–0.068) for Hg, for As 0.0595 g/kg/day (range: 0.001–0.755), for Pb 0.0235 g/kg/day (range: 5.401–0.232) and for Cd 0.009 g/kg/day (range: 1.322 × 10^−5^–0.860). The species *Musa paradisiaca* presented the lowest EDI values for Hg, Pb, Cd, and Pb for Medio Atrato. Only the species *Alibertia patinoi* (Medio Atrato) and *Ocimum campechianum* (Bojayá) exceeded the RfDo value limit for As (As 0.30 g/kg/day) (Table 1).

For fish, the mean EDI values were 1.374 μg/kg/day (range: 0.122–3.556) for Hg, for As 0.119 g/kg/day (range: 0.016–0.672), for Pb 0.067 μg/kg/day (range: 0.015–0.067) and, for Cd, standard values of 0.003 g/kg/day were shown for all municipalities (Table 1). The species *Ctenolucius beani* presented the highest EDI values for Hg in the municipalities of Murindó (3.612 μg/kg/day), Bojayá (3.556 μg/kg/day), Medio Atrato (1.370 μg/kg/day) and Vigía del Fuerte. (1.655 μg/kg/day), followed by *Hoplias malabaricus* in Murindó (2.589 μg/kg/day), Bojayá (1.579 μg/kg/day), Medio Atrato (1.719 μg/kg/day), Vigía del Fuerte (1.694 μg/kg/day) and *Trachelyopterus fisheri* in Vigía del Fuerte (3.028 μg/kg/day), Murindó (2.673 μg/kg/day) and Medio Atrato (1.513 μg/kg/day). All the other species reported in this study presented values above what is recommended. However, species such as *Andinoacara pulcher* in Murindó and Vigía del Fuerte, *Leporinus muyscorum* in Murindó, *Hypostomus hondae* in Vigía del Fuerte, presented RfDo values below that established for Hg (0.16 μg/kg/day). In the case of As, the EDI values were above the established threshold (As 0.30 μg/kg/day) in species such as *H. hondae* (Murindó), and *Geophagus Pellegrini* (Vigía del Fuerte). In general, the EDI values for Pb remained below the limits; however, the species *L. muyscorum* in Murindó presented values above the RfDo limits (0.390 μg/kg/day) (Table 2).

### 3.5. Assessment of Non-carcinogenic Health Risk

To establish the carcinogenic risk in the population exposed to the consumption of fish, fruit-bulbs, and vegetables-stems, the target hazard quotients (TQH and TTQH) were calculated. When the THQ value is less than one, it means that the exposure level is less than the RfDo, indicating that daily exposure at this level is unlikely to cause adverse effects during a person’s lifetime [33,47]. For the present evaluation, the mean TQH values of fish were 3.700 for Hg (range: 0.643–21.668), for Cd 0.023 (range: 0.022–0.023), for As 0.228 (range: 0.053–1.398) and for Pb 0.012 (range: 0.004–0.140).

The THQ value for fish consumption for the metals Cd and Pb was less than one for all fish species in all the municipalities studied. Most of the fish species presented a TQH for Hg higher than one; the highest values were reported in *A. pardalis* (21.688) in Vigía del Fuerte, *C. beani* (21.649 and 21.313) in Murindó and Bojayá, and *H. malabaricus* (10.30) in Medio Atrato. However, species such as *Andinoacara pulcher* and *L. muyscuorum*, in the municipalities of Murindó and Vigía del Fuerte, and *H. hondae* in Murindó presented TQH values for Hg lower than one. For As, the TQH index was higher than one for *H. hondae* and *T. fisheri* in Medio Atrato; for Cd and Pb all THQ values were less than one. The relative contributions to the total THQ score showed that Hg was the main contributor to risk. The average THQ for Hg far exceeded one (6.768). In general, the species with the highest values of EDI, THQ, and TTHQ for Hg were those of carnivorous, piscivorous and omnivorous habits with a tendency to carnivory including *H. malabaricus*, *A. pardalis*, *R. quelen*, *T. fisheri*, *S. aequilabiatus*, *C. kraussii*, *C. atratoensis*, *C. beani* and *P. schultzi* (Table 1).

For fruit and vegetables, the mean TQH values were 0.074 (range: 0.001–0.412) for Hg, for Cd 0.054 (range: 7.924 × 10^−5^–1.322), for As 0.017 (range: 0.001–0.207) and for Pb 0.006 (range: 1.479 × 10^−5^–0.063). The TQH values for vegetable consumption, based on the mean concentrations of Hg, Pb, and As for all vegetable species, were generally less than one in the four municipalities studied, except in species such as *Manihot esculenta* (1.322) in Murindó and *Colocasia esculenta* (2.405) in Bojayá. The relative contributions to total THQ showed that Cd was the main element contributing to the risk for vegetables (Table 1).

### 3.6. Assessment of Carcinogenic Health Risk

The calculation of the carcinogenic risk was applied only to the metals As and Pb due to their carcinogenic effect on humans. It was applied to the species of fish and vegetables commonly consumed by the inhabitants of the middle basin of the Atrato River. To do this, the USEPA has recommended using the carcinogenic risk index (CR). If the CR values are less than 1.0 × 10^−6^, they are considered negligible, while a CR value greater than 1.0 × 10^−4^ indicates potential adverse effects in humans [37,38]. The data obtained for vegetables contaminated with As and Pb in the groups of fruit-bulbs and vegetables-stems collected in the municipalities of Medio Atrato, Bojayá, Vigía del Fuerte, and Murindó showed values lower than 1.0 × 10^−6^ (Table 1). Therefore, our results indicate that there was no carcinogenic risk to the health of the population from the consumption of these foods.

However, with respect to the carcinogenicity values of the metals Pb and As, the results showed, for all the species captured in the middle basin of the Atrato River, values lower than 1.0 × 10^−6^ for Pb. However, the results of the evaluation of the carcinogenic health risk of As through fish consumption showed that the CR values for the fish species *H. malabaricus*, *T. fisheri*, *P. schultzi*, *C. beani*, *Caquetaia umbrifera*, *Sternopygus aequilabiatus*, *A. fasciatus*, *L. muyscorum*, *Prochilodus magdalenae*, *H. hondae* in Medio Atrato; *A. pardalis*, *Geophagus pellegrini*, *Caquetaia kraussii*, *Rhamdia quelen*, *L. muyscorum*, *P. magdalenae*, *H. hondae* in Vigía del Fuerte; and *A. pardalis*, *H. malabaricus*, *C. beani*, *R. quelen*, *L. muyscorum*, *P. magdalenae*, *H. hondae* in Murindó were above the 1.0 × 10^−6^ limit, but below the 1.0 × 10^−4^ limit (Table 1). The results of this study indicate that there is a significant carcinogenic risk for the health of the inhabitants of the middle basin of the Atrtao River, mainly due to As, through the consumption of fish.

In general, the results of EDI, THQ, TTHQ, and CR show that the inhabitants of the middle basin of the Atrato River could present health problems during their lives, due to the consumption of fish, depending on the type of fish they consume.

**Table 1 ijerph-20-00435-t001:** Estimated daily intake (EDI), estimated target hazard quotients (THQ), total estimated hazard quotient (TTHQ) and carcinogenic risk of the most consumed vegetables (fruits and vegetables) for individual metals from vegetables consumption.

Fish Species	Hg		Cd		As		Pb			As *	Pb *
	EDI^+^	THQ	EDI^+^	THQ	EDI^+^	THQ	EDI^+^	THQ	TTHQ	CR	CR
**Medio Atrato**											
**Fruits-Tubers**											
*Colocasia esculenta*	0.021	0.128	0.008	0.052	0.077	0.021	0.232	0.063	0.264	6.356 × 10^−7^	1.894 × 10^−6^
*Oryza sativa*	0.021	0.129	0.011	0.070	0.051	0.014	0.007	0.002	0.215	4.199 × 10^−7^	6.324 × 10^−8^
*Alibertia patinoi*	0.006	0.041	0.001	0.011	**0.356**	0.097	0.007	0.002	0.151	2.908 × 10^−6^	6.198 × 10^−8^
*Musa × paradisiaca*	0.013	0.080	0.006	0.036	0.018	0.004	0.024	0.006	0.126	1.486 × 10^−7^	2.023 × 10^−7^
*Musa balbisiana*	0.012	0.073	0.090	0.543	0.024	0.006	0.039	0.010	0.632	6.467 × 10^−5^	2.026 × 10^−7^
*Cocos nucifera*	0.035	0.021	0.008	0.049	0.007	0.002	0.012	0.003	0.075	6.333 × 10^−8^	1.012 × 10^−7^
*Citrus aurantifolia*	0.001	0.004	Nd	Nd	Nd	Nd	Nd	Nd	0.004	Nd	Nd
*Musa sp*	0.027	0.163	0.009	0.055	0.109	0.029	0.092	0.025	0.272	8.903 × 10^−7^	7.577 × 10^−7^
*Musa sapientum*	0.019	0.117	0.006	0.039	0.062	0.017	0.027	0.007	0.180	5.073 × 10^−7^	2.206 × 10^−7^
*Solanum sessilliflorum*	0.014	0.084	0.034	0.208	0.011	0.003	0.018	0.005	0.300	9.500 × 10^−8^	1.518 × 10^−7^
*Zea mays*	0.011	0.067	0.003	0.018	0.109	0.029	0.012	0.003	0.117	8.918 × 10^−7^	1.012 × 10^−7^
*Dioscorea trifida*	0.020	0.122	0.007	0.045	0.019	0.005	0.030	0.008	0.180	1.567 × 10^−7^	2.504 × 10^−7^
*Carica papaya*	0.004	0.302	0.040	0.243	0.034	0.009	0.034	0.009	0.563	2.795 × 10^−7^	2.851 × 10^−7^
*Manihot esculenta*	0.012	0.074	0.006	0.038	0.016	0.004	0.026	0.007	0.123	1.330 × 10^−7^	2.215 × 10^−7^
**Vegetables-Stems**											
*Eryngium foetidum*	0.015	0.090	0.007	0.042	0.014	0.054	0.028	0.007	0.193	4.447 × 10^−7^	2.285 × 10^−7^
*Ocimum campechianum*	0.047	0.028	0.022	0.132	0.221	0.060	0.126	0.034	0.254	1.806 × 10^−6^	1.032 × 10^−6^
*Origanum vulgare*	0.018	0.110	0.010	0.065	0.063	0.017	0.028	0.007	0.199	5.143 × 10^−7^	2.338 × 10^−7^
Saccharum officinalis	0.015	0.094	Nd	Nd	Nd	Nd	Nd	Nd	0.094	Nd	Nd
**Bojayá**											
**Fruits-tubers**											
*Colocasia esculenta*	0.014	0.088	0.401	**2.405**	0.231	0.063	0.034	0.009	**2.565**	1.887 × 10^−6^	2.782 × 10^−7^
*Musa balbisiana*	0.013	0.081	0.037	0.225	0.023	0.006	0.038	0.010	0.322	1.948 × 10^−7^	3.112 × 10^−7^
*Manihot esculenta*	0.013	0.083	0.012	0.075	0.075	0.020	0.025	0.006	0.184	6.166 × 10^−7^	2.042 × 10^−7^
*Musa × paradisiaca*	0.011	0.068	0.019	0.114	0.099	0.027	0.023	0.006	0.215	8.137 × 10^−7^	1.945 × 10^−7^
*Citrus aurantifolia*	0.002	0.013	0.860	0.476	0.093	0.025	0.008	0.002	0.516	7.591 × 10^−7^	6.566 × 10^−8^
*Alibertia patinoi*	0.003	0.018	0.001	0.010	0.011	0.003	0.007	0.002	0.033	9.258 × 10^−8^	5.958 × 10^−8^
*Musa sp*	0.022	0.136	0.008	0.053	0.022	0.006	0.036	0.009	0.204	1.856 × 10^−7^	2.967 × 10^−7^
*Solanum sessilliflorum*	0.010	0.065	0.033	0.203	0.011	0.003	0.017	0.004	0.275	9.138 × 10^−8^	1.459 × 10^−7^
*Zingiber officinale*	0.004	0.024	0.005	0.034	0.022	0.006	0.001	0.001	0.065	1.873 × 10^−7^	1.605 × 10^−8^
**Vigía del fuerte**											
**Fruits-tubers**											
*Colocasia esculenta*	0.019	0.118	Nd	Nd	Nd	Nd	Nd	Nd	0.118	Nd	Nd
*Oryza sativa*	0.002	0.015	0.001	0.011	0.057	0.015	0.007	0.002	0.043	4.653 × 10^−7^	6.191 × 10^−8^
*Citrus aurantifolia*	0.001	0.003	Nd	Nd	Nd	Nd	Nd	Nd	0.003	Nd	Nd
*Musa balbisiana*	0.015	0.090	0.009	0.057	0.024	0.006	0.038	0.010	0.163	1.984 × 10^−7^	3.170 × 10^−7^
*Zea mays*	0.005	0.032	0.002	0.017	0.075	0.020	0.012	0.003	0.072	6.120 × 10^−7^	9.906 × 10^−8^
*Musa sp*	0.014	0.087	0.009	0.054	0.107	0.029	0.037	0.026	0.196	8.757 × 10^−7^	7.747 × 10^−7^
*Musa × paradisiaca*	0.012	0.073	0.005	0.035	0.215	0.059	0.024	0.006	0.173	1.759 × 10^−6^	1.981 × 10^−7^
*Manihot esculenta*	0.019	0.116	0.033	0.199	0.170	0.046	0.025	0.006	0.367	1.390 × 10^−6^	2.080 × 10^−7^
*Solanum sessilliflorum*	0.007	0.047	0.020	0.121	0.070	0.019	0.029	0.008	0.195	5.717 × 10^−7^	2.426 × 10^−7^
**Vegetables-Stems**											
*Ocimum campechianum*	0.029	0.176	0.024	0.148	0.199	0.054	0.008	0.002	0.380	1.626 × 10^−6^	6.934 × 10^−8^
*Eryngium foetidum*	0.006	0.040	0.004	0.028	0.052	0.014	0.003	0.001	0.083	4.309 × 10^−7^	2.476 × 10^−8^
*Origanum vulgare*	0.007	0.043	0.005	0.031	0.051	0.014	0.006	0.001	0.089	4.173 × 10^−7^	1.486 × 10^−8^
*Ocimum basilicum* L.	0.015	0.092	0.002	0.012	0.175	0.048	0.042	0.011	0.163	1.434 × 10^−6^	3.443 × 10^−7^
*Minthostachys mollis*	0.006	0.041	Nd	Nd	Nd	Nd	Nd	Nd	0.041	Nd	Nd
*Basella rubra var*	0.009	0.055	0.001	0.005	0.012	0.003	0.003	0.001	0.064	1.012 × 10^−7^	2.972 × 10^−8^
*Saccharum officinalis*	0.014	0.085	Nd	Nd	Nd	Nd	Nd	Nd	0.085	Nd	Nd
**Murindó**											
**Fruits-tubers**											
*Colocasia esculenta*	0.032	0.192	0.138	0.832	0.251	0.068	0.036	0.009	1.101	2.047 × 10^−6^	2.962 × 10^−7^
*Musa balbisiana*	0.068	0.412	0.009	0.059	0.064	0.017	0.040	0.012	0.500	5.274 × 10^−7^	3.314 × 10^−7^
*Musa sapientum*	0.029	0.174	0.006	0.040	0.062	0.017	0.027	0.007	0.238	5.100 × 10^−7^	2.258 × 10^−7^
*Musa × paradisiaca*	0.016	0.099	0.011	0.068	0.231	0.063	0.025	0.006	0.236	1.885 × 10^−6^	2.071 × 10^−7^
*Manihot esculenta*	0.024	0.147	0.220	1.322	0.122	0.033	0.026	0.007	1.509	9.967 × 10^−7^	2.175 × 10^−7^
*Citrus aurantifolia*	0.001	0.003	Nd	Nd	Nd	Nd	Nd	Nd	0.003	Nd	Nd
*Alibertia patinoi*	0.003	0.022	0.006	0.037	0.011	0.003	0.007	0.002	0.064	9. 679 × 10^−8^	6.344 × 10^−8^
*Zea mays*	0.006	0.041	0.004	0.028	0.077	0.021	0.012	0.003	0.093	6.322 × 10^−7^	1.035 × 10^−7^
**Vegetables-stems**											
*Eryngium foetidum*	0.007	0.043	0.002	0.012	0.057	0.015	0.003	0.001	0.071	4.661 × 10^−7^	2.589 × 10^−8^
*Ocimum basilicum* L.	0.014	0.085	Nd	Nd	Nd	Nd	Nd	Nd	0.085	Nd	Nd
*Minthostachys mollis*	0.014	0.084	0.001	0.004	0.258	0.070	0.003	0.001	0.159	2.110 × 10^−6^	2.589 × 10^−8^
*Ocimum tenuiflorum*	0.020	0.123	0.002	0.013	Nd	Nd	0.008	0.002	0.138	Nd	7.251 × 10^−8^
*Allium fistulosum* L.	8.324 × 10^−5^	0.001	1.322 × 10^−5^	7.924 × 10^−5^	0.001	0.001	5.401 × 10^−5^	1.479 × 10^−5^	9.405 × 10^−5^	3.937 × 10^−9^	4.402 × 10^−10^
*Saccharum officinalis*	0.008	0.053	Nd	Nd	Nd	Nd	Nd	Nd	0.053	Nd	Nd

The order of the municipalities is from upstream to downstream on the Atrato River. * Carcinogenic risk. EDI^+^ values in (μg/kg/day). Oral reference dose (RfD) values used were: Hg = 0.16 μg/kg/day; Cd = 1.00 μg/kg/day; Pb = 3.50 μg/kg/day; and As = 0.30 μg/kg/day. The values highlighted in bold are above the established limits.

**Table 2 ijerph-20-00435-t002:** Estimated daily intake (EDI), estimated target hazard quotients (THQ), total estimated hazard quotient (TTHQ) and carcinogenic risk of the most consumed fish for individual metals from fish consumption.

Fish Species	Hg		Cd		As		Pb			As *	Pb *
	EDI+	THQ	EDI+	THQ	EDI+	THQ	EDI+	THQ	TTHQ	CR	CR
**Medio Atrato**											
*Hoplias malabaricus*	**1.719**	**10.30**	0.003	0.023	0.117	0.376	0.074	0.020	10.719	**1.695 × 10^−3^**	6.057 × 10^−7^
*Trachelyopterus fisheri*	**1.513**	**9.070**	0.003	0.023	0.223	**1.340**	0.047	0.013	10.446	**3.217 × 10^−3^**	3.890 × 10^−7^
*Pseudopimelodus schultzi*	**1.622**	**9.722**	0.003	0.023	0.062	0.200	0.045	0.012	9.957	9.003 × 10^−5^	3.703 × 10^−7^
*Ctenolucius beani*	**1.370**	**8.215**	0.003	0.023	0.156	0.501	0.015	0.004	8.743	**2.256 × 10^−3^**	1.295 × 10^−7^
*Caquetaia umbrifera*	**0.401**	**2.407**	0.003	0.023	0.169	0.541	0.058	0.016	2.987	**2.436 × 10^−3^**	4.777 × 10^−7^
*Sternopygus aequilabiatus*	**0.617**	**3.700**	0.003	0.023	0.237	0.757	0.024	0.006	4.486	**3.409 × 10^−3^**	2.003 × 10^−7^
*Rhamdia quelen*	**0.608**	**3.646**	0.003	0.023	0.067	0.214	0.060	0.016	3.899	9.674 × 10^−5^	4.958 × 10^−7^
*Astyanax fasciatus*	**0.781**	**4.681**	0.003	0.023	0.172	0.550	0.015	0.004	5.258	**2.476 × 10^−3^**	1.295 × 10^−7^
*Andinoacara pulcher*	**0.522**	**3.129**	0.003	0.023	0.041	0.133	0.064	0.017	3.302	6.003 × 10^−5^	5.256 × 10^−7^
*Leporinus muyscorum*	**0.431**	**2.585**	0.003	0.023	0.155	0.497	0.051	0.014	3.119	**2.237 × 10^−3^**	4.207 × 10^−7^
*Prochilodus magdalenae*	**0.292**	**1.752**	0.003	0.023	0.140	0.450	0.034	0.009	2.234	**2.026 × 10^−3^**	2.797 × 10^−7^
*Hypostomus hondae*	**0.487**	**2.922**	0.003	0.023	**0.437**	**1.398**	0.082	0.022	4.365	**6.291 × 10^−3^**	6.723 × 10^−7^
**Bojayá**											
*Hoplias malabaricus*	**1.579**	**9.465**	0.003	0.022	0.063	0.201	0.090	0.024	9.532	9.070 × 10^−5^	7.374 × 10^−7^
*Ctenolucius beani*	**3.556**	**21.313**	0.003	0.022	0.025	0.080	0.015	0.004	21.419	3.610 × 10^−5^	1.245 × 10^−7^
*Prochilodus magdalenae*	**0.501**	**3.003**	0.003	0.022	0.056	0.179	0.057	0.015	3.219	8.090 × 10^−5^	4.697 × 10^−7^
**Vigía del fuerte**											
*Ageneiosus pardalis*	**3.615**	**21.668**	0.003	0.022	0.120	0.384	0.049	0.013	22.087	**1.728 × 10^−3^**	4.033 × 10^−7^
*Hoplias malabaricus*	**1.694**	**10.155**	0.003	0.022	0.042	0.136	0.015	0.004	10.317	6.131 × 10^−5^	1.268 × 10^−7^
*Trachelyopterus fisheri*	**3.028**	**18.149**	0.003	0.022	0.009	0.031	0.004	0.004	18.206	1.400 × 10^−5^	1.268 × 10^−7^
*Pseudopimelodus schultzi*	**2.105**	**12.619**	0.003	0.022	0.043	0.139	0.050	0.013	12.793	6.272 × 10^−5^	4.076 × 10^−7^
*Ctenolucius beani*	**1.655**	**9.923**	0.003	0.022	0.044	0.143	0.026	0.007	10.095	6.441 × 10^−5^	2.132 × 10^−7^
*Sternopygus aequilabiatus*	**2.776**	**16.641**	0.003	0.022	0.039	0.126	0.015	0.004	16.793	5.705 × 10^−5^	1.268 × 10^−7^
*Cynopotamus atratoensis*	**2.621**	**15.710**	0.003	0.022	0.044	0.143	0.015	0.004	15.879	6.467 × 10^−5^	1.268 × 10^−7^
*Geophagus Pellegrini*	**0.943**	**5.653**	0.003	0.022	**0.672**	**2.149**	0.154	0.042	7.866	**9.674 × 10^−3^**	1.256 × 10^−6^
*Caquetaia kraussii*	**1.446**	**8.667**	0.003	0.022	0.072	0.230	0.015	0.004	8.923	**1.036 × 10^−3^**	1.268 × 10^−7^
*Rhamdia quelen*	**1.639**	**9.824**	0.003	0.022	0.187	0.598	0.015	0.004	10.448	**2.694 × 10^−3^**	1.268 × 10^−7^
*Pimelodus sp*	**0.531**	**3.185**	0.003	0.022	0.052	0.167	0.032	0.008	3.382	7.536 × 10^−5^	2.632 × 10^−7^
*Andinoacara pulcher*	0.149	0.893	0.003	0.022	0.068	0.217	0.039	0.010	1.142	9.788 × 10^−5^	3.251 × 10^−7^
*Leporinus muyscorum*	**0.347**	**2.080**	0.003	0.022	0.209	0.668	0.144	0.039	2.809	**3.006 × 10^−3^**	1.179 × 10^−6^
*Prochilodus magdalenae*	**0.347**	**2.083**	0.003	0.022	0.097	0.310	0.015	0.004	2.419	**1.398 × 10^−3^**	1.268 × 10^−7^
*Hypostomus hondae*	0.107	0.643	0.003	0.022	0.079	0.253	0.015	0.004	0.922	**1.142 × 10^−3^**	1.268 × 10^−7^
**Murindó**											
*Ageneiosus pardalis*	**2.633**	**15.781**	0.003	0.023	0.100	0.321	0.064	0.017	16.142	**1.448 × 10^−3^**	5.244 × 10^−7^
*Hoplias malabaricus*	**2.589**	**15.521**	0.003	0.023	0.082	0.262	0.045	0.012	15.818	**1.182 × 10^−3^**	3.712 × 10^−7^
*Trachelyopterus fisheri*	**2.673**	**16.025**	0.003	0.023	0.016	0.053	0.038	0.010	16.111	2.409 × 10^−5^	3.117 × 10^−7^
*Pseudopimelodus schultzi*	**1.310**	**7.855**	0.003	0.023	0.022	0.071	0.016	0.004	7.953	3.197 × 10^−5^	1.325 × 10^−7^
*Ctenolucius beani*	**3.612**	**21.649**	0.003	0.023	0.071	0.228	0.016	0.004	21.904	**1.028 × 10^−3^**	1.325 × 10^−7^
*Sternopygus aequilabiatus*	**2.468**	**14.796**	0.003	0.023	0.010	0.032	0.057	0.015	14.866	1.464 × 10^−5^	4.704 × 10^−7^
*Caquetaia kraussii*	**1.904**	**11.412**	0.003	0.023	0.037	0.119	0.091	0.025	11.579	5.383 × 10^−5^	7.443 × 10^−7^
*Rhamdia quelen*	**1.970**	**11.808**	0.003	0.023	0.077	0.247	0.069	0.019	12.097	**1.113 × 10^−3^**	5.633 × 10^−7^
*Astyanax fasciatus*	**0.318**	**1.908**	0.003	0.023	0.060	0.193	0.158	0.043	2.167	8.729 × 10^−5^	1.293 × 10^−6^
*Pimelodus punctatus*	**0.726**	**4.356**	0.003	0.023	0.030	0.096	0.016	0.004	4.479	4.328 × 10^−5^	1.325 × 10^−7^
*Pimelodella chagresi*	**0.443**	**2.657**	0.003	0.023	0.023	0.075	0.016	0.004	2.759	3.382 × 10^−5^	1.325 × 10^−7^
*Andinoacara pulcher*	0.123	0.740	0.003	0.023	0.029	0.093	0.513	0.140	0.996	4.195 × 10^−5^	4.183 × 10^−6^
*Leporinus muyscorum*	0.122	0.736	0.003	0.023	0.106	0.341	0.390	0.106	1.206	**1.535 × 10^−3^**	3.181 × 10^−6^
*Prochilodus magdalenae*	**0.468**	**2.807**	0.003	0.023	0.091	0.293	0.080	0.021	3.144	**1.322 × 10^−3^**	6.537 × 10^−7^
*Hypostomus hondae*	**0.165**	0.994	0.003	0.023	**0.312**	0.997	0.044	0.012	2.006	**4.490 × 10^−3^**	3.619 × 10^−7^

The order of the municipalities is from upstream to downstream on the Atrato River. * Carcinogenic risk. EDI^+^ values in (μg/kg/day). Oral reference dose (RfD) values used are: Hg = 0.16 μg/kg/day; Cd = 1.00 μg/kg/day; Pb = 3.50 μg/kg/day; and As = 0.30 μg/kg/day. The values highlighted in bold are above the established limits.

### 3.7. Risk Assessment by MeHg in Most Consumed Fish

The mean concentrations of MeHg (μg kg^−1^) and the percentage of MeHg (%MeHg) in the fish species consumed by the inhabitants of the middle basin of the Atrato River are presented in Appendix A. Of the total fish species studied, 221 individuals exceeded the limit for populations at risk, which was established at 200 µg kg^−1^ of MeHg [41]. Among these 102 individuals exceeded the maximum recommended limit for human consumption, established at 500 µg kg^−1^ of MeHg (Appendix A). All the municipalities studied, except Medio Atrato, reported concentrations that exceeded the permissible limits of 500 µg kg^−1^. The species with the highest concentrations were: in Murindó, *C. beani*, *A. pardalis*, *T. fisheri*, *H. malabaricus*, *S. aequilabiatus*, *R. quelen*, *C. kraussii*; in Bojaya, *C. beani*, *A. pardalis*, *T. fisheri*, *C. atratoensis*; and in Vigía del Fuerte, *P. schultzi*. Other species, such as *P. schultzi* (Murindó and Medio Atrato), *C. kraussii* (Vigia del Fuerte), *H. malabaricus* (Bojayá, Medio Atrato, and Vigía del Fuerte), *T. fisheri*, *A. fasciatus*, *G. Pellegrini* (Medio Atrato), and *C. beani* (Medio Atrato and Vigía del Fuerte) exceeded the limit for vulnerable populations (WCHA) of 200 µg kg^−1^. In general, the carnivorous species with the highest concentrations of MeHg was *A. pardalis* with 956.82 μg kg^−1^ in the municipality of Vigía del Fuerte, which represents an important species in the food security of the population under study. However, for the species *P. magdalenae*, the highest consumption for which was reported in the middle basin of the Atrato river, the concentrations of MeHg did not exceed any of the thresholds established by the WHO (200 μg kg^−1^ and 500 μg kg^−1^) (Appendix A).

When the daily intake rate (RI) was estimated for the fish species commonly consumed in the studied sites, six species presented values higher than 500 g/week and lower than 700 g/week (which is of particular relevance when assessing vulnerable populations, e.g., children and women of childbearing age), including *C. atratoensis* (Vigía del Fuerte), *S. aequilabiatus*, *C. umbrifera* (Medio Atrato), *T. fisheri* (Medio Atrato and Murindó), *P. punctatus* (Murindó) and *C. beani* (Murindó and Bojayá). In the municipality of Murindó, the species *R. quelen*, *L. muyscorum* and *P. schultzi*, presented the highest levels of IR, with consumption levels of 1254.4 g/week, 1433.6 g/week and 1587.2 g/week, respectively (Appendix A). Regarding the frequency of consumption (FIR), for the species *P. magdalenae* weekly consumption of four or more times was reported in all the municipalities studied; however, the municipality of Murindó presented the highest FIR values for *P. schultzi* (6.2 days/week), *L. muyscorum* (5.6 days/week), and *R. quelen* (4.9 days/week). Values close to this limit were observed in *R. quelen* (2.7 days/week) and *T. fisheri* (2.7 days/week), both in the municipality of Vigía del Fuerte (Appendix A). In relation to the estimated weekly intake (EWI), the results showed that 10 fish species exceeded the potential weekly intake threshold (PTWI) for the GP group (3.2 μg kg bw/week). For the WCHA group (1.6 μg kg bw/week), it was also shown that the species with piscivorous, carnivorous and omnivorous habits with a carnivorous tendency, had the highest EWI values. These species were *A. pardalis* (Vigía del Fuerte and Murindó), *C. beani* (Medio Atrato, Bojayá, and Murindó) and *R. quelen* (Murindó) with values of 4.7, 2.8 and 2.7 times the PTWI for the GP group and 9.1, 5.4 and 5.3 times the PTWI for the WCHA group, respectively. The results showed that fish consumption limits higher than those recommended (MFW) were obtained in all the municipalities studied. In Medio Atrato, the species presented values between 0.3 to 1.6 and 0.5 to 3.3 times the PTWI for the GP and WCHA group, respectively; in Bojayá, between 0.6 to 2.1 and 1.2 to 4.4 times, respectively; in Vigía del Fuerte between 0.1 to 9.1 and 0.2 to 9.1 times, respectively; and in the municipality of Murindó between 0.1 to 3.0 and 0.2 to 6.0 times, respectively. The lowest values recorded for *Hypostomus hondae* (Vigía del Fuerte) were about 10.3- and 5.5-fold lower than the PTWI for the GP and WCHA, respectively. High consumption species, such as *A. pardalis*, *H. malabaricus*, *C. beani* and *P. schultzi* represent a serious risk to the health of the inhabitants in the studied areas due to the high concentrations of MeHg in their tissues, such that it is recommended for riverside populations to reduce or eliminate the consumption of these fish. Similarly, species of high consumption preference, such as *P. magdalenae*, presented values 2.7 times below the recommended consumption in Medio Atrato, 2.6 times below in Vigía del Fuerte, 1.7 times below in Bojayá, and 1.5 below times in Murindó, and for *L. muyscorum*, 5.4 times below in the municipality of Medio Atrato. These results suggest that these species can be consumed frequently by the inhabitants of the studied areas as their content does not exceed recommended MFW consumption limits. In addition, these species could be important for replacement of species that present high concentrations in the diet.

### 3.8. Diagnosis of the Population

The contamination index (Pi) [40] was used to show the degree of contamination with Hg for each species of fish in each municipality, taking as a reference the permissible limits established by the WHO (500 µg kg ^−1^ and 200 µg kg ^−1^) [47,48]. Appendix A shows that, when the Pi values were calculated according to the WHO limit [42], the fish species *A. pardalis* in the municipalities of Murindó and Vigía del Fuerte, *T. fisheri* in Vigía del Fuerte and Murindó, *C. beani* in Bojayá, and *C. atratoensis* in Vigía del Fuerte presented slight contamination (1< Pi ≤ 2) (Appendix A). The contamination index was also calculated based on the WHO threshold [47] (2008) indicating that *P. schultzi* in Murindó, *H. malabaricus* in Bojayá and Vigía del Fuerte, *C. beani* and *T. fisheri* in Medio Atrato, and *C. kraussii* and *G. Pellegrini* in Vigía del Fuerte showed a slight degree of contamination (1< Pi ≤ 2) (Appendix A). The species *Sternopygus aequilabiatus* (Murindó), *A. pardalis* (Murindó), *R. quelen* (Murindó and Vigía del Fuerte), *C. kraussii* (Medio Atrato), *Hoplias malabaricus* (Murindó and Medio Atrato), *P. schultzi* (Medio Atrato and Vigia del Fuerte) and *C. beani* and *R. quelen* (Vigia del Fuerte) showed a moderate degree of contamination (2 < Pi < 3). Similarly, *T. fisheri* in Murindó and Vigia del Fuerte, *C. beani* in Murindó and Bojayá, and *S. aequilabiatus*, *C. atratoensis* and *A. pardalis* in Vigia del Fuerte, presented a high degree of contamination (Appendix A).

## 4. Discussion

The middle basin of the Atrato river of the Colombian Pacific is within an area where the Murindó, Bebará, Bebaramá, and Neguá rivers flow—these tributaries are associated with high levels of gold-mining activity. This area has suffered significant impacts on the rivers and surrounding soils due to indiscriminate gold mining, resulting in contamination with significant concentrations of heavy metals of the fish in the area which are important for food security. This contamination has also affected the crops that play an important role in the food security of the inhabitants of this area. However, there are no records of metal concentrations in this type of food, nor of any risk assessments. Based on ruling T-622 and the Minamata agreement, it is important to assess the risk to human health from the consumption of food contaminated with these heavy metals by the inhabitants of the middle basin of the Atrato River.

In this study, the concentrations of metals in fruits-tubers and vegetable-stems followed the order As > Pb > Hg > Cd. In Medio Atrato, fruits-tubers, in general, presented low levels of metals, except for As in the species *Alibertia patinoi* (203.16) and Zea mays (38.16). Vegetable-stems showed higher levels of As, especially *Ocimum campechianum* (110.41), *Eryngium foetidum* (76.12), and *Origanum vulgare* (146.72); in Bojaya, Vigia del Fuerte, and Murindó, the metal concentrations were found to be the same. It is important to highlight that the concentrations of As in vegetables in all the municipalities studied were between 3.18–391.75 mg/kg, which is well above the maximum standards established internationally for vegetables (0.5 mg/kg) [45,46]. These results are similar to those observed in studies carried out on *Daucus carota*, *Cynara scolymus*, and *Petroselinum crispum* in the city of Sibaté (Colombia) [48].

The concentrations of As in these foods could be derived from the original soil material or the application of fertilizers and pesticides still used in agricultural activities [49,50,51]. In addition, studies have reported that there are differences in the concentrations in vegetables. Leafy vegetables are a group of plants recognized for having a high capacity for heavy metal accumulation [51,52], with plant species having different capacities for the absorption and accumulation of metal(oids), associated with factors such as the different characteristics of the soil or the growth period of each plant [53,54]. It is evident that these plants have good translocation characteristics for As, either derived from mining waste sources from soil removal in gold extraction activities or from natural sources in the Earth’s crust [51]. The concentrations of Hg in vegetables-stems were at higher levels compared to fruit-tubers; this accumulation behavior by this type of plant species has been reported in research carried out in China [52]. However, the concentrations of Hg reported in the municipality of Lloró for vegetables were found to be lower than those in fruits and tubers [51]. These concentrations could be related to low translocation factors and common mining processes in the studied areas, with atmospheric deposition of Hg not having a significant influence on crops due to high precipitation in these areas, causing Hg to be deposited on the ground [51,55]. Pb concentrations in fruits and vegetables from the middle Atrato basin did not exceed WHO permissible consumption limits [41,42]. These results are in agreement with studies carried out on fruits and vegetables from the African countries of South Africa and Mozambique [14].

Pb levels were higher in the vegetables-stems, consistent with the findings of investigations carried out on vegetables in Baiyin, China [56]. The municipalities of Vigía del Fuerte and Medio Atrato reported higher concentration levels in vegetables-stems above those allowed by the Codex [45,46], especially for *E. foetidum* (cilantro), *O. campechianum* (basil), *O. vulgare* (oregano), *B. rubra var* (spinach), *M. mollis* (pennyroyal) and *O. basilicum L*. (white basil) with concentrations between 4.33–66.71 mg/kg. These Pb concentrations in vegetables, in general, showed values above the maximum standards established internationally by the Codex for vegetables (0.1 mg/kg) in all the studied municipalities [45,46]. The concentration levels observed in our study were very high compared to those carried out in Arequipa (Peru) in quinoa, corn and rice products, with high Pb concentrations observed of 0.55, 0.75 and 5.08 mg kg^−1^, respectively [57]. They are consistent with Pb concentrations reported in cabbage (23.1 ± 1.5), lettuce (17.2 ± 2.7), and tomato (15.0 ± 1.1) from the city of Arba Minch (Ethiopia) [58] and concentrations of between 0.84 and 12.5 for vegetables of dietary importance in the city of Sibaté (Colombia) [48]. In the case of fruit-tubers, such as *C. esculenta* (28.34 ± 0.9), Pb levels in Medio Atrato were high, contrary to the results reported for concentrations in this tuber grown in the Canary Islands (Spain) [59].The above is possibly related to the growth of these species in contaminated soils as a result of soil removal for mining activities and the use of irrigation water contaminated with metals, something which is very common in the studied areas of the Atrato environment [60]. Another reason could be the high density of stomata in the leaves of these plants, which allows for the accumulation of atmospheric Pb [61]. Pb can be adsorbed and fixed in the clay material of soils. It is characterized by being a highly mobile metal, with mobility increasing with pH, particularly in acid conditions, such as those found in the soils of the Middle Atrato, which could favor its adsorption and accumulation [62]. In addition, the results show that these plants have the potential and capacity to bioaccumulate and translocate concentrations of Pb, as well as the other metals evaluated in this study. The levels of Cd in the fruits-bulbs of the municipalities of Murindó and Bojayá were high compared to the other two municipalities, particularly for *C. esculenta*, which is frequently consumed by the inhabitants, which presented concentrations of 1.06–50.92 mg/kg, exceeding the Codex limit [45,46]. However, the levels of Pb reported in *C. esculenta* cultivated in the Canary Islands (Spain) showed levels below the Codex limits [59]. These plants are characterized by being tubers; the bulbs grow underground and are in direct contact with the contaminated soil present in the study area and, as a result, can accumulate significant concentrations of Cd. There are reports of the large capacity for accumulation of heavy metals from this type of plant because its parts grow under the earth’s surface comprising large tubers for consumption [63,64]. In general, Cd, like other metals, can be absorbed by the pores of the stomata of the leaves. However, unlike other toxic metals, such as Pb, it has high mobility in the soil, is easily absorbed by the roots, and transported to the shoots, and is uniformly distributed in the plants [64]. Its high degree of bioaccumulation is due more to soil contamination than to atmospheric deposition [65]. In our research, the concentrations of Cd did not exceed the permissible limits of consumption specified by the WHO [41,42], in contrast to the findings of Genthe et al. [14] for fruits and vegetables from African countries. In addition, the bioaccumulation ranges observed were above those reported by Real et al. [66], in which the established range was 0.003–1.616 for *O. sativa*. On the other hand, the data from our research showed concentrations higher than those reported for *Solanum lycopersicum* [58], *Daucus carota*, *Cynara escolymus* and *Petroselinum crispum* [48], *M. paradisiaca* and *C. aurantifolia* [67], *Apium graveolens*, *Lepidium sativum* and *Porrum de Alliuml* [68], and *Z. mays* and *O. sativa* [57]. Subsistence agriculture for the inhabitants of the Atrato river basin has developed on the ground and the river is the main source of water irrigation for crops. However, the high impact of gold mining in the Atrato river basin and its tributaries has generated high levels of contamination, which has contributed to the contamination of crops with Hg, Cd, Pb, and As. Therefore, these anthropogenic activities constitute an imminent risk to human health in the riverside populations of the basin. The estimated daily intake rates (EDI) of fruits-tubers and vegetables-stems for all the inhabitants of the middle basin of the Atrato river are shown in Table 1, as well as data for the average body weight by population group, the intake by food category, and the reference doses (RfDs) for Hg, Cd, As, and Pb [47]. In general, the EDI values for none of the vegetable groups exceeded the oral RfD for metals in the studied areas, except for the *Alibertia patinoi* fruit in Medio Atrato for As. Therefore, exposure to Hg, Cd, As, and Pb through the consumption of fruits-tubers and vegetables-stems does not represent a threat to the health of the inhabitants of the municipalities of the middle basin of the Atrato River since the calculated EDI values do not exceed the reference dose tolerable intake values (RfD: 0.16 μg Hg/kg/day, 1.0 μg Cd/kg/day, 0.3 μg As/kg/day, 3.5 μg Pb/kg/day) [47]. Our results are in contrast to results obtained for fruits and tubers in the municipality of Lloró (Chocó-Colombia) by Marrugo-Madrid et al. [51], where the exposure of As through the consumption of fruits could represent a threat to the health of all population groups studied since the calculated EDI values exceed the tolerable consumption reference dose by 10 times.

The non-carcinogenic health risk was also evaluated based on THQ. With a THQ < 1, the exposed population should not experience any adverse risk, but if THQ > 1, the population could experience health risks not related to cancer. Table 1 shows that most of the THQ values did not exceed one, except for *Colocasia esculenta* (HQ = 2.405) in the municipality of Bojayá for As. These results indicate that, in general, people would not experience significant health risks from the ingestion of individual meta(loids) through vegetable consumption. Carcinogenic risk (CR) evaluation was also carried out for As and Pb using the same method. For this investigation values greater than 1.0 × 10^−4^ were taken as indicators of risk for vegetables [37]. In our study, none of the plant species evaluated presented a carcinogenic risk for As and Pb. These results were in contrast to those of similar investigations of vegetables showing carcinogenic risk associated with Pb and As concentrations in Bangladesh and Ireland [66,69], and observations from Peru, where some species, such as *O. sativa*, exposed to As showed evidence of possible risk to the population [57]. Similarly, in the municipality of Lloró (Colombia), the fruits and tubers showed a possible carcinogenic risk by exposure to As. However, in all the municipalities, the studied vegetables presented levels above the Codex. Therefore, considering the accumulative properties of heavy metals, these may represent a health hazard for the riverside populations of the middle basin of the Atrato River.

The results showed that the concentrations in fish followed the order Hg > As > Pb > Cd, with high concentration levels of Hg, MeHg, and As observed. *C. beani* presented the highest concentrations of Hg (1008.0 ± 552.7 g kg^−1^) in the municipality of Bojayá. However, the municipality of Vigía del Fuerte had a greater number of species with high concentrations, including *C. kraussii*, *R. quelen*, *C. beani*, *H. malabaricus*, *P. schultzi*, *C. atratoensis*, *S. aequilabiatus*, *T. fisheri*, and *A. pardalis*. These areas are characterized by extensive mining activities on the Murrí River and other tributaries [21]. In the Atrato river basin, other studies reported similar data for *A. pardalis, H. malabaricus*, and *C. beani*; the observed concentrations were related to extractive mining of the Atrato river and its tributaries [9,21]. The high concentrations of THg found were consistent with observations by Vargas-Licona and Marrugo-Negrete [70] who warned about the toxicological risk due to mining in some ecosystems in Colombia. The concentrations of MeHg in fish from the middle basin of the Atrato, especially those with a carnivorous habit, exceeded the permissible safe consumption limits by 500 μg kg^−1^ [42]. These results were similar to the findings of Salazar-Camacho et al. [21] in this area of the basin. The concentrations of THg and MeHg in fish reported in this investigation were similar to those reported in other investigations in the Atrato river basin [8,9,21]. These results are also consistent with other investigations, where species of carnivorous habit c-p and oc, such as *H. malabaricus*, *C. kraussii,* and *A. pardalis*, presented greater bioaccumulation of THg and MeHg [71].

The mean concentrations of Pb and Cd in fish were 12.03 ± 24.4 and 1.1 ± 2.2 g kg^−1^, respectively. These concentrations were higher than those for investigations of fish from the Buriganga River (Bangladesh), specifically, the species *Heteropneustes fossilis*, *Channa striata*, *Labeo rohita*, and *Catla catla* [66]; however, other studies have reported lower concentrations in fish from the Ciénaga Grande de Santa Marta (Colombia) [18].

Concentrations of the metals Pb and Cd for the fish species in the middle basin exceeded the limits established by the Codex [45,46]; however, they did not exceed the permissible intake limits defined by the WHO [41,42]. In general, the concentrations of As in fish for the basin were above what is allowed [45,46]. The municipalities of Medio Atrato and Vigia del Fuerte recorded the highest levels of average concentrations in the basin (134.2 ± 61.0 and 187.2 ± 164.8 ug kg^−1^). The species *P. magdalenae* and *Leporinus muyscorum*, which are of gastronomic importance, did not exceed the permissible intake limits defined by the WHO [41,42], which is a reflection of the preferred feeding habit of the species in the ecosystem. The EDI of fish for all the inhabitants of the middle basin of the Atrato river are shown in Table 2, as well as the intake by food category, and the RfDs for Hg, Cd, As, and Pb [38]. In general, the EDI values for Hg and As for the vegetable groups exceeded the oral RfD for metals in the studied areas, especially for Hg, where the values were between 0.107–3.615; only four fish species had concentrations below the RfD. The EDI values for As exceeded the RfD only for *H. hondae* (Medio Atrato and Mutindó) and *G. Pellegrini* (Vigia del Fuerte). Therefore, the exposure to Hg and As through the consumption of fish could represent a threat to the health of all the inhabitants of the municipalities of the middle basin of the Atrato River since the calculated EDI values exceeded the reference dose tolerable intake. Our results were similar to the findings of studies of fish in the municipality of Lloró (Chocó-Colombia), where the exposure to As and Hg through the consumption of fish could represent a threat to the health of all the studied population groups, especially for Hg [51].

The TQH data for Hg in our study were very high 0.736–21.68, which represents a risk for the coastal populations through consumption of fish contaminated with Hg. These data were similar to those reported for the municipalities of Vigía del Fuerte, Murindó, Bojayá, Medio Atrato [8], and Lloró [51]. Other studies have shown that species such as *P. schultzi*, *A. pardalis*, *S. aequilabiatus*, *R. quelen*, *H. malabaricus*, *Cathorops melanopus*, *Centrpomo undecimalis*, *C. umbrifera*, *C. kraussii*, *Prochilodus* sp., *Prochilodus punctatus*, *Prochilodus magdalenae*, *Spatuloricaria atratoensis*, *Leporinus muyscorum*, *Hemiancistrus wilsoni*, and *Cyphochara Magdalena* presented values greater than one for the THQ index [9]. Our results showed (Table 2) that the CR values through As exposure from fish consumption were well above 1 × 10^−4^ for all the municipalities (except Bojayá), especially in Medio Atrato (CR: 9.674 × 10^−5^–6.291 × 10^−3^), indicating that the local population should reduce its intake of fish. Species such as *A. pardalis, P. magdalenae, H. malabaricus, L. muyscorum*, and *R. quelen*, which are of gastronomic importance, showed levels of contamination of carcinogenic importance for As, except for the municipality of Bojayá. Salazar-Camacho et al. [8], reported similar data for carcinogenic risk through As exposure. The Colombian National Institute of Cancerology (INC) has estimated that there are about 74.8 new cases of cancer per 100,000 inhabitants, although in the Pacific Region the increase in national carcinogenic risk is not considered to be decisive; however, heavy metal contamination and excessive intake of contaminated food may affect local and national carcinogenic risk in the future. Genes are influenced by the environment and their modification can lead to many types of cancer—there is a direct relationship between environmental contaminants and the increase in many types of cancer.

MeHg is the most toxic form of Hg and exposure to this pollutant is associated with the consumption of fish. In this study, it represented between 71.08–99.21% of the THg. Therefore, it is important to assess the potential risk of exposure to MeHg in the population based on the estimated weekly intake (EWI), corresponding to the maximum quantity of fish that can be consumed weekly (MFW) per person without harmful effects on health, and the permissible safety level concentration of MeHg in fish for human consumption. Worryingly, our study showed that the WCHA group had a fish intake with a frequency from 0.2 to 9.1 times more than recommended, compared to the 1.3–2.1 times reported by Salazar-Camacho et al. [8] who also presented MFW values higher than those recommended. This report shows that women of childbearing age or pregnant women (WCHA group), especially in the municipalities of Medio Atrato, Vigía del Fuerte, and Munrindo, were at risk of having high concentrations of MeHg in placental tissue, blood, and cord blood, which could affect the health of the mother, fetus, and newborn.

Appendix A shows that the highest EWI corresponded to *A. pardalis* in Vigía del Fuerte (10.65 μg/kg/week). According to the recommended limits for the consumption (MFW) of fish, in the municipalities of Medio Atrato, Bojayá, Vigía del Fuerte, and Murindó, consumption of species such as *C. beani*, *P. schultzi*, *H. malabaricus*, *T. fisheri*, *A. pardalis*, *S. aequilabiatus*, *R. quelen*, *C. atratoensis*, and *C. kraussii* exceeded the weekly recommended intake by residents. Our results agree with previous reports for some of these fish species in the Atrato river basin [8,51]. The inhabitants (including children, pregnant women, and women of childbearing age, the elderly, and adults) of all the studied areas presented EWI values higher than the reference (PTWI) and the estimated values (MFW), respectively. As such, there is a potential risk to the health of these inhabitants. The results of this study are important because they show the impact of fish consumption on the most vulnerable population (WCHA) of the middle basin of the Atrato River. Local and national authorities need to implement strategies to prevent children and the WCHA group from consuming fish containing high levels of Hg, such as *H. malabaricus*, *A. pardalis*, *P. schultzi*, *C. kraussi* and *R. quelen*. They should, alternatively, recommend that these groups eat fish with low concentrations of MeHg, such as *Andinoacara pulcher*, *Leporinus muyscorum*, and *Hypostomus hondae*. Therefore, it is recommended to establish continuous monitoring of the content of metal(oids) in the riverside populations of the basin and their food, and to implement bioremediation strategies to decontaminate the soil and water in these areas, to guarantee the consumption of safe foods with respect to heavy metal content and improve the health security of the inhabitants who depend on the aquatic resources and crops of the area as the basis of food security and economic sustenance.

## 5. Conclusions

High concentrations of As, Hg, Pb, and Cd were identified in fish, fruits-tubers and vegetables-stems commonly consumed by inhabitants of the middle basin of the Atrato River, which exceeded Codex limits. Similarly, in fish, the concentrations of MeHg and THg exceeded the limits established by the WHO/FAO for vulnerable populations and the rest of the adult population, especially for carnivorous fish species.

A high carcinogenic and non-carcinogenic risk was evidenced for the inhabitants of the middle basin of the Atrato River due to the consumption of fish contaminated with high concentrations of As, MeHg, and THg. However, the risk associated with consumption of vegetables was very low with only a couple of species showing some degree of risk for As.

Health risks from consuming MeHg-contaminated fish are a matter of concern since many fish species were consumed at levels exceeding the recommended weekly intake (MFW) and the PTWI for all population groups, in all areas studied. Thus, it is recommended that the consumption of carnivorous species is reduced or replaced, and that consumption of non-carnivorous species, such as *P. magdalenae*, occurs instead.

The combined exposure to the four metals through the consumption of fish, fruits, and vegetables would probably result in exceeding the RfD for the population of the middle Atrato basin. It is important that further studies of multiple exposure to toxins found in the foods most consumed by the inhabitants are undertaken. In addition, it is necessary that periodic monitoring of heavy metals is carried out in riverside populations and their food and that bioremediation strategies for soils and water sources are implemented to reduce the concentration of these pollutants and to improve food production and quality.

## Figures and Tables

**Figure 1 ijerph-20-00435-f001:**
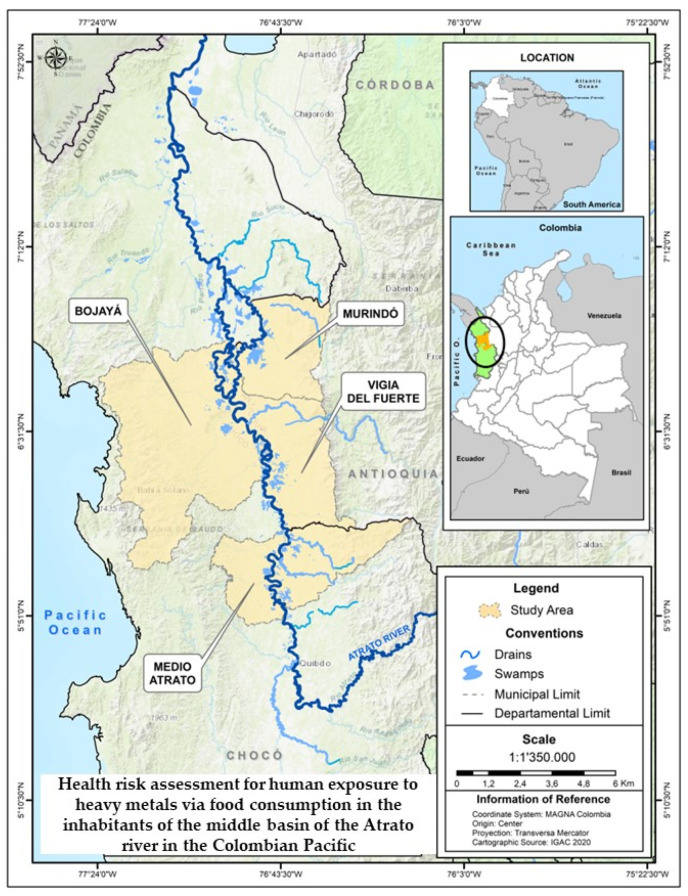
Map of the four municipalities studied belonging to the middle basin of the Atrato River (Chocó-Colombia): Medio Atrato, Bojaya, Vigía del Fuerte, and Murindó.

## Data Availability

In this study, the data are for the exclusive use of the Ministry of Science, Technology and Innovation of Colombia and no data was reported in the study.

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
