# Peer review of "Health Risk Assessment for Human Exposure to Heavy Metals via Food Consumption in Inhabitants of Middle Basin of the Atrato River in the Colombian Pacific"

_ijerph, 2022, doi:10.3390/ijerph20010435_

Round 1

Reviewer 1 Report

I would like to congratulate the authors for this wonderful paper.

The amount of data generated is enormeous and the risk characterization has been very well conducted.

More importantly, the authors manage to suggest risk managing solutions and actions.

There are some minor questions:

Why using the Codex limit and the limits established by the 21 WHO/FAO, especially carnivorous fish species?. Are there not Colombian limits? Could this be a suggestion for Colombiasn Regulators?

Line 237, the PTWI reference must be added

table 1: "(weekly/days)" is not clear. Please modify the units used to describe the intake

Table 2: the scientific name must be followed by the (N) in the title of that first column

Why didn`t authors estimate a mean value for each group of samples and with so generate a more clear understanding of the situation of each family product. The same for the EDI, why not estimating a mean EDI from each family of products?

For the discussion, I suggest to consider the following article:

Luis-González G, Rubio C, Gutiérrez Á, González-Weller D, Revert C, Hardisson A. Essential and toxic metals in taros (Colocasia esculenta) cultivated in the Canary Islands (Spain): evaluation of content and estimate of daily intake. Environ Monit Assess. 2015 Jan;187(1):4138. doi: 10.1007/s10661-014-4138-2. Epub 2014 Nov 21. PMID: 25412891.

Reviewer 2 Report

This study was designed to evaluate heavy metal pollution in fish, fruits, and vegetables and human exposure in the riverside inhabitants of the middle basin of the Atrato river, where is highly impacted by gold mining. Generally, the workload is very large and the authors collected so many diet samples and determined 4 typical heavy metals. However, the novelty is low and the used method is very common. I think it may be accepted in IJERPH after major revision.

Specific comments:

1. 154 samples of different fruits and vegetables and 440 samples of fish were analyzed by atomic absorption spectroscopy. Why the authors did not collect rice or wheat flour samples? It’s the staple food with large daily consumed quantity. The authors should check the diet structure.

2. The authors used DORM-2 as CRM for Hg analysis. The THg concentrations in CRM DORM-2 standard of dog-fish muscle certificate averaged at 4.47 ± 0.32 µg g-1, which were much higher than the actual samples. For instance, the minimum values of Hg were found between two municipalities, in Vigía del Fuerte, Leporinus muyscorum with 44.5±22.7 μg kg-1; and in Murindó with species Hypostomus hondae (41.5±32.5 μg kg-1) and Andinoacara pulcher (32.9±4.3 μg kg-1). As well, in Line 159, the unit was not shown.

3. Figure 3. Concentrations of Hg(3a), As(3b), Pb(3c), and Cd(3d) in fruits and vegetables of common consumption in municipalities of the middle basin of the Atrato River. The unit of Hg in 1st figure was not right.

4. The authors did not consider the speciation of As. For instance, the inorganic As is more toxic. In fish samples, the organic As is the main form.

5. Organization of the paper. There are too many big tables in the manuscript. I suggested the authors move some to the SI, such as the tables with detail information fruits, vegetables, and fish. The paper is too long and the authors should simplify it to make it more easily for read.

Round 2

Reviewer 2 Report

The authors made the revisions according to the review comments. As I raised before, the paper is too long, and the authors should simplify it. There are too many big tables in the manuscript. I suggested the authors move some to the SI. Please consider this.

Author Response

  1. 154 samples of different fruits and vegetables and 440 samples of fish were analyzed by atomic absorption spectroscopy. Why the authors did not collect rice or wheat flour samples? It’s the staple food with large daily consumed quantity. The authors should check the diet structure.

Response: Thanks for the observation, rice and wheat flour in this middle area of the Atrato River are bought in stores, people consume these products from different commercial brands, in this area these products are not grown, and on this research we rely in foods that were grown in the study area

  1. The authors used DORM-2 as CRM for Hg analysis. The THg concentrations in CRM DORM-2 standard of dog-fish muscle certificate averaged at 4.47 ± 0.32 µg g-1, which were much higher than the actual samples. For instance, the minimum values of Hg were found between two municipalities, in Vigía del Fuerte, Leporinus muyscorumwith 44.5±22.7 μg kg-1; and in Murindó with species Hypostomus hondae(41.5±32.5 μg kg-1) and Andinoacara pulcher (32.9±4.3 μg kg-1). As well, in Line 159, the unit was not shown.

Response: we share this suggestion, thanks The mercury concentration of this certified reference material (CRM DORM-2 standard of dog-fish muscle) is relatively high but it is almost all organomercury. We have made many measurements with this standard and we have always had similar and reliable results. Corretions in line 155,159,161,162

  1. Figure 3. Concentrations of Hg(3a), As(3b), Pb(3c), and Cd(3d) in fruits and vegetables of common consumption in municipalities of the middle basin of the Atrato River. The unit of Hg in 1stfigure was not right.

Response: Thanks for the observation, the unit is in mg/kg, in figure 3a in the unit that is for the concentration of Hg, so I still don't see the error

  1. The authors did not consider the speciation of As. For instance, the inorganic As is more toxic. In fish samples, the organic As is the main form.

Response: The results surprised us, we did not expect significant concentrations of arsenic, but in future studies we will consider speciation for Arsenic

  1. Organization of the paper. There are too many big tables in the manuscript. I suggested the authors move some to the SI, such as the tables with detail information fruits, vegetables, and fish. The paper is too long and the authors should simplify it to make it more easily for read.

Response: we share this suggestion, thanks

Comments and Suggestions for Authors

The authors made the revisions according to the review comments. As I raised before, the paper is too long, and the authors should simplify it. There are too many big tables in the manuscript. I suggested the authors move some to the SI. Please consider this.

Response: The observations have been made, the manuscript was simplified with only two tables and three figures, a total of five tables were placed in the supplementary material, in addition minor editions were made in English and corrections were made in the organization of tables and in the scientific names of tables. species
